# Three-dimensional printing of photonic colloidal glasses into objects with isotropic structural color

Ahmet F. Demirörs [1] ✉, Erik Poloni [1,6], Maddalena Chiesa [1], Fabio L. Bargardi[1], Marco R. Binelli [1], Wilhelm Woigk[1], Lucas D. C. de Castro [1,2,7], Nicole Kleger[1], Fergal B. Coulter [1], Alba Sicher [3], Henning Galinski [4], Frank Scheffold [5] & André R. Studart [1] ✉

Structural color is frequently exploited by living organisms for biological functions and has also been translated into synthetic materials as a more durable and less hazardous alternative to conventional pigments. Additive manufacturing approaches were recently exploited for the fabrication of exquisite photonic objects, but the angle-dependence observed limits a broader application of structural color in synthetic systems. Here, we propose a manufacturing platform for the 3D printing of complex-shaped objects that display isotropic structural color generated from photonic colloidal glasses. Structurally colored objects are printed from aqueous colloidal inks containing monodisperse silica particles, carbon black, and a gel-forming copolymer. Rheology and Small-Angle-X-Ray-Scattering measurements are performed to identify the processing conditions leading to printed objects with tunable structural colors. Multimaterial printing is eventually used to create complex-shaped objects with multiple structural colors using silica and carbon as abundant and sustainable building blocks.

Living organisms and geological processes give rise to vivid colors in nature by forming structures with characteristic length scales comparable to the wavelength of visible light. This optical effect, known as structural color, can be found in iridescent gemstones called opals[1–4] as well as in butterfly wings[5], beetles[6], and fish[7,8]. In these photonic materials, diffraction gratings and periodically ordered structures generate color through constructive interference of specific wavelengths of light[9–12] and may be combined with light-absorbing pigments[13]. Because they enable light manipulation in specific directions, these natural structures have inspired the development of a broad range of synthetic photonic materials[14,15]. The possibility to produce non-bleaching colors using sustainable chemistries also makes such nanostructured materials attractive alternatives to harmful pigment-based colorants[16,17].

A particular feature of some of the above-mentioned natural photonic materials is the dependence of color on the illumination angle, which leads to the characteristic iridescent effect. While this angle dependence can be exploited to create band-gap photonic structures for light manipulation, it also limits the suitability of structural color as an alternative to the isotropic coloration generated by absorption-based pigments. Notably, nature also offers examples of scattering from isotropic nanostructures that exhibit structural color

[1]Complex Materials, Department of Materials, ETH Zurich, 8093 Zurich, Switzerland. [2]Federal University of São Carlos, Department of Materials Engineering, São Carlos, SP, Brazil. [3]Laboratory for Soft and Living Materials, Department of Materials, ETH Zurich, 8093 Zurich, Switzerland. [4]Laboratory for Nanometallurgy, Department of Materials, ETH Zurich, 8093 Zurich, Switzerland. [5]Soft Matter and Photonics, Department of Physics, University of Fribourg, 1700 Fribourg, Switzerland. [6]Present address: High Enthalpy Flow Diagnostics Group, Institute of Space Systems, University of Stuttgart, 70569 Stuttgart, Germany. [7]Present address: São Carlos Institute of Physics, University of São Paulo, 13566-590 São Carlos, SP, Brazil. ✉e-mail: ahmet.demiroers@mat.ethz.ch; andre.studart@mat.ethz.ch

with little angle-dependence[12,18–21]. This has motivated the development of bio-inspired materials with isotropic structural color[22–28]. Photonic glasses created through the self-assembly of colloidal particles has received significant attention in this regard[24,29–34]. Such materials combine the long-range disorder needed to generate isotropic properties with the structural periodicity desired for the emergence of structural color. Mutually repulsive colloidal particles self-assemble into glass structures if the volume fraction of particles is increased above approximately 0.55[32,35]. To prevent crystallization, slightly polydisperse colloids are often employed. Angle-independent structural color has been demonstrated in thin films of colloidal glasses made from particles with controlled polydispersity[22]. The use of a broadband absorber to suppress angle-independent multiple scattering has enabled the preparation of isotropic structural color also in thick films[18,31,32,36].

Despite the enticing prospects of creating structural color using self-assembling colloidal glasses, current examples are still limited to the two-dimensional and simple geometries generated by casting, coating, and pressing processes[36–40]. To partially fill this gap, additive manufacturing technologies have recently been exploited to produce three-dimensional photonic objects with intricate geometries and increased shape complexity[38,39,41–43]. Photonic periodic structures have been fabricated by two-photon polymerization lithography of shrinking and shape-memory polymers[42,43]. In other examples, tailored copolymers have been used to self-assemble into photonic crystals during fused filament fabrication[41] or direct-writing of polymer solutions[38]. The direct-writing approach has also been applied to assemble monodisperse particles into colorful colloidal crystals[39] and disordered structures[44]. However, current additive manufacturing methods have been restricted either to 2D demonstrations[44] or to 3D crystalline structures with angle-dependent photonic properties[39], preventing a broader application of structural color as alternative to conventional pigments and dyes.

Here, we report a 3D printing platform for the manufacturing of complex-shaped objects with angle-independent structural color. The formation of isotropic structural color relies on the self-assembly of monodisperse particles into photonic colloidal glasses with tunable local order. Colloidal particles are printed into three-dimensional objects using the direct ink writing (DIW) approach. To design printable inks with programmed structural color, we first study the viscoelastic properties of pastes containing monodisperse silica particles, carbon black, and rheology modifiers. Next, the processing conditions required for the formation of photonic colloidal glasses within the printed object are systematically evaluated. Finally, we demonstrate the potential of the printing platform by fabricating three-dimensional objects with isotropic multiple colors arising from the controlled formation of photonic colloidal glasses.

## Results and discussion

### Printing platform and the ink formulation

The DIW of objects with isotropic structural color relies on the design of inks that simultaneously enable 3D shaping and self-assembly of colloidal glasses (Fig. 1). Shaping of objects with bespoke color patterns occurs via the top-down deposition of inks of specific compositions using the programmable spatial control provided by the 3D printer. The formation of a colloidal glass with tailored photonic properties is driven by the bottom-up self-assembly of particles. Fulfilling the requirements of the printing and assembly processes is not straightforward. On the one hand, top-down printing through the DIW technique demands a colloidal ink with rheological properties that allow for the extrusion of filaments into distortion-free printed structures[45,46]. On the other hand, the bottom-up assembly of particles into a colloidal glass depends on the formation of a highly packed arrangement of particles without the onset of crystallization. The volume fraction of particles in the ink plays a crucial role in the above printing and assembly processes. For DIW, the volume fraction of particles is typically high but should still be lower than the maximum packing density to allow the ink to flow during extrusion through the printing nozzle. In contrast, the formation of a colloidal glass is favored when particles are jammed in a disordered arrangement by bringing their volume fraction close to the packing density when a glass transition is observed.

To meet these opposing demands, we designed a water-based colloidal ink with a volume fraction of particles that is initially low enough to enable DIW but eventually reaches the maximum packing limit desired for glass formation when the liquid and gel phase of the ink is removed by heat treatment. In addition to the color-forming silica particles, the ink also contains a gel-forming agent and carbon black nanoparticles. A temperature-responsive triblock copolymer of polypropylene oxide-polyethylene oxide-polypropylene oxide (PEO-PPO-PEO) is used as the gel-forming agent to tune the rheological behavior of the ink. The carbon black particles are utilized as an absorption medium between the silica particles to reduce multiple scattering of the incoming light[18]. Finally, polyacrylic acid is used as a dispersant for the electrosteric stabilization of the colloidal particles at neutral to high pH.

Distinct formulations were investigated in order to develop inks with the rheological properties needed for DIW and with the ability to

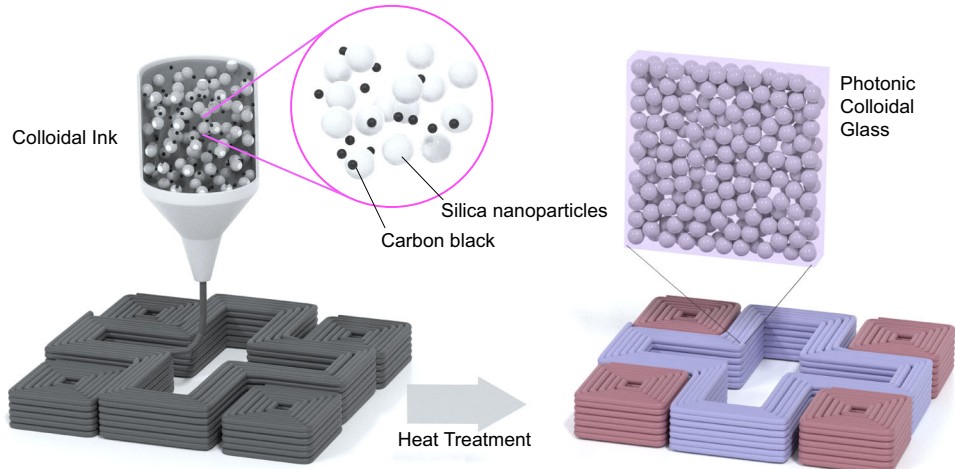

Colloidal Ink

Silica nanoparticles

Carbon black

Photonic Colloidal Glass

Heat Treatment

**Fig. 1 | The printing platform.** Schematics illustrating the 3D printing of colloidal inks into objects with isotropic structural color. Coloration is generated by photonic colloidal glasses obtained upon complete drying of the as-printed objects.

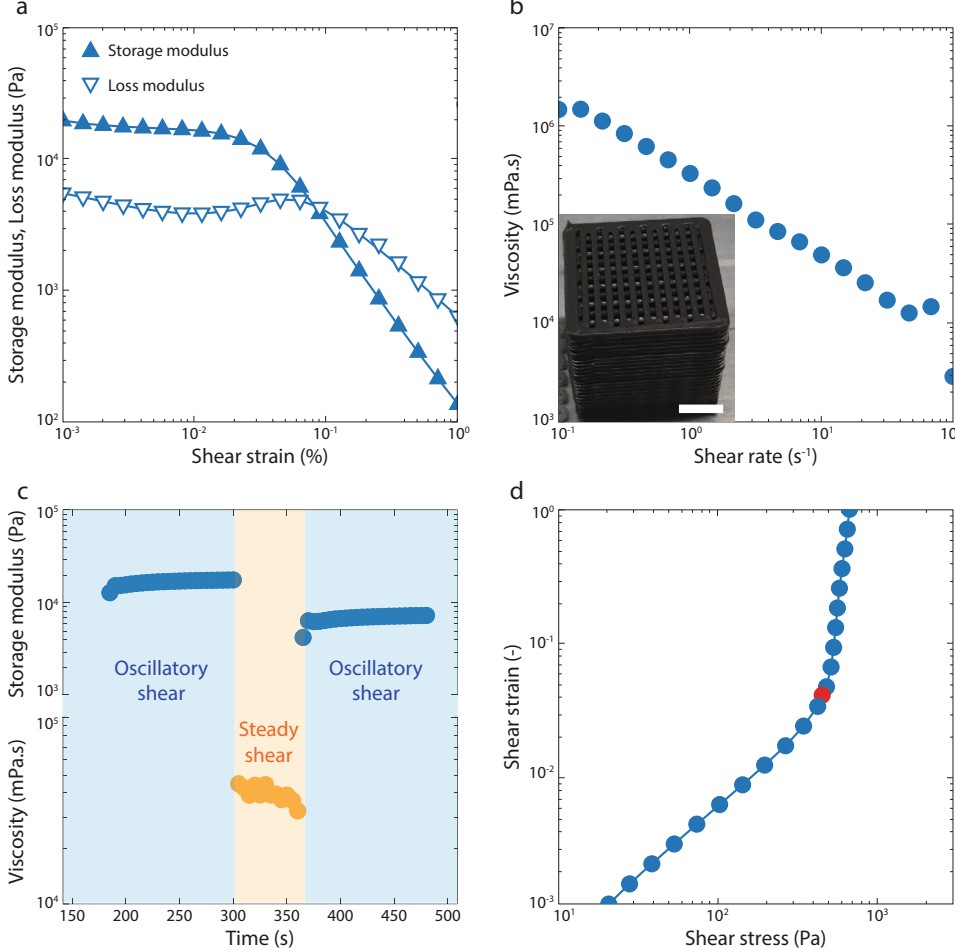

**Fig. 2 | Rheology of a representative colloidal ink. a** Storage ($G'$) and loss ($G''$) moduli of the ink as a function of the magnitude of the applied oscillatory shear strain. **b** Apparent viscosity of the ink as a function of the shear rate applied in a steady-shear experiment. The inset shows an as-printed object with a grid-like architecture. Scale bar: 5 mm. **c** Elastic recovery experiments that simulate the printing process through a sequence of oscillatory, steady-shear, and oscillatory measurements. The oscillatory tests are performed before and after the application of a print-emulating steady shear at a rate of 10 s$^{-1}$. **d** Shear strain of the ink when subjected to an increasing shear stress in a quasi-static experiment. The red full circle indicates the apparent yield stress of the ink. The colloidal ink contained 32 vol% of 250 nm silica particles suspended in an aqueous solution with 19 wt% PEO-PPO-PEO copolymer and 0.7 wt% carbon black. Source data for plots are provided as a Source Data file.

generate photonic colloidal glasses upon heat treatment. Because the structural color of the final object is directly affected by the characteristic length scale of the colloidal glass, ink formulations with silica particles of different sizes were prepared and studied. Preliminary printing experiments led to the identification of an ink formulation with promising rheological properties. The selected formulation contained monodisperse 250 nm silica particles at a volume fraction ($\phi$) of 0.32 in an aqueous solution with 28 wt% PEO-PPO-PEO copolymer (Pluronic F-108) and up to 3 wt% carbon black.

**Rheology of the Ink**

The viscoelastic and flow properties of the selected ink were evaluated by a series of rheological measurements and matched the requirements for distortion-free DIW (Fig. 2). The manufacturing of complex objects by DIW typically requires that inks display high storage modulus ($G'$), shear-thinning behavior, high yield stress ($\tau_y$) and fast elastic recovery[45,46]. Taking a typical grid-like architecture as an exemplary complex-shaped object, we estimate the storage modulus and yield stress levels needed to obtain distortion-free structures with well-defined geometry (Fig. 2a, d and Supplementary Note 1). A high storage modulus is desired to prevent sagging of free-spanning filaments[45], whereas a high yield stress is required to prevent

distortion of the printed structure due to capillary forces[46]. While these properties ensure mechanical stability of the final printed object, shear-thinning behavior at stresses beyond the yield stress and quick elastic recovery facilitate extrusion and promote gelation of the ink during and shortly after extrusion of the filaments, respectively.

Oscillatory experiments indicate that such an ink shows a clear viscoelastic response, which is characterized by a gel-like elastic behavior at low strains ($G' > G''$) and fluidization above a critical shear strain ($G' < G''$, Fig. 2a). The storage modulus ($G'$) of 18.8 kPa obtained for the optimized ink is sufficient to print grid-like structures with negligible sagging of spanning filaments. On the basis of simple beam theory, we expect grids printed with such an ink to show minimum distortion as long as the spanning length does not exceed 9.2 times the diameter of the filament (Supplementary Note 1). In terms of yield stress, shear experiments reveal that the optimized ink exhibits an $\tau_y$ value of 441 Pa, which allows for printing of structures with local radii of curvature down to 43 μm without capillary-driven distortions. The optimized ink was also found to be shear-thinning within the range of shear rates expected during printing (Fig. 2b). Finally, shearing experiments designed to emulate the extrusion process reveal that 45 % of the ink's storage modulus recovers within seconds after deposition of a filament (Fig. 2c). These rheological properties

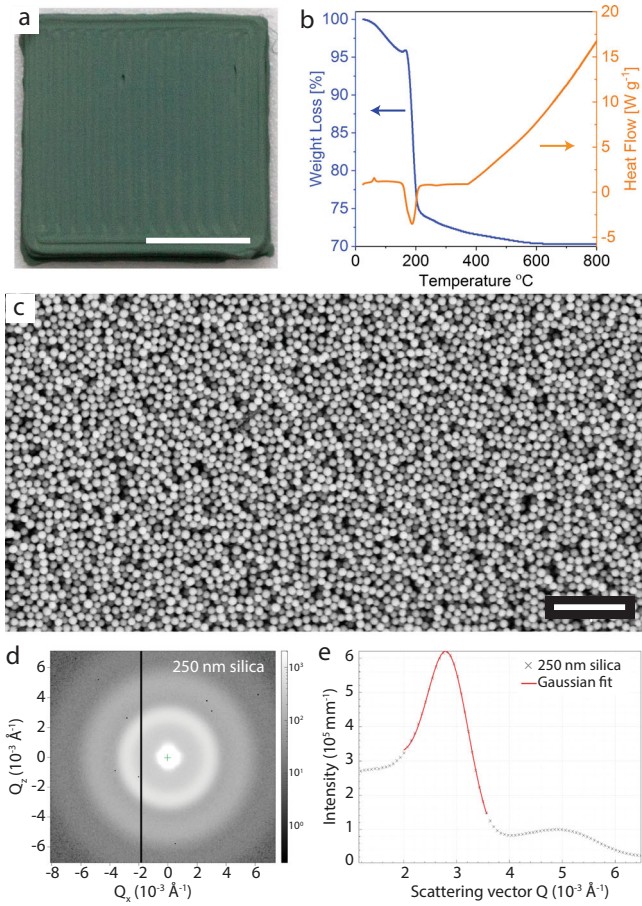

**Fig. 3 | Structure of printed and heat-treated photonic colloidal glass. a** Picture of a printed object after heat treating at 200 °C. **b** Thermal gravimetric analysis (TGA) and differential thermal analysis (DTA) of an as-printed object subjected to heating in air. **c** Scanning electron microscopy (SEM) image, **d** SAXS pattern, and **e** azimuthal X-Ray intensity as a function of scattering vector (Q) obtained for a printed colloidal glass after heat treating at 200 °C. All structures were printed from a colloidal ink containing 250 nm silica particles. Scale bar in **a** is 5 mm and in **c** is 2 μm. The vertical line in (d) is an instrumentation artifact due to the interface between detectors (see Supplementary Note 4 for details). Source data for **b** and **e** are provided as a Source Data file.

eventually allowed for the 3D printing of large grid-like objects using the optimized ink formulation (Fig. 2d). Notably, the as-printed object is black, indicating that its color is dominated by the highly-absorbing carbon particles present in the ink.

**Structural and thermal characterization of the printed samples**
Drying of the as-printed object at 25 °C was found to change the color of the structure from black to gray. Surprisingly, further heat treatment at the higher temperature of 200 °C led to a strong and vivid green color (Fig. 3a). The fact that the color is angle-independent suggests that a photonic colloidal glass with length scales on the order of the wavelengths of visible light was formed upon drying at this higher temperature. To better understand the origin of the observed color, we first performed thermogravimetric analysis on partially dried as-printed specimens (Fig. 3b). The results show that the as-printed sample loses only 5% of its initial mass when heated up to 150 °C, whereas a more substantial mass loss of 20% is observed upon further heating to 200 °C. TGA measurements performed on the individual components of the ink (see Supplementary Note 2 and Supplementary Fig. 1) indicates that the mass loss up to 150 °C corresponds to the removal of the remaining water that is bound to the

copolymer-rich continuous phase between silica particles. Our experiments suggest that the decomposition of the gel-forming PEO-PPO-PEO copolymer is the main cause for the 20% mass loss at 200 °C. While the copolymer alone was found to decompose only at around 300 °C, the swelling of these hydrophilic molecules after mixing with water and their possible adhesion to the particle surfaces are expected to decrease the stability of the PEO-PPO-PEO chains and shift their disintegration temperature to 200 °C (see Supplementary Note 2 and Supplementary Fig. 1).

Scanning electron microscopy of specimen heat treated at 200 °C reveals that the removal of water and copolymer from the as-printed material leads to a glass-like microstructure of densely packed silica particles without long-range order (Fig. 3c). The colloidal glass obtained indicates that the random configuration of particles in the ink and in the room-temperature-dried object is preserved during the heat treatment process at 200 °C (see Supplementary Fig. 2). In contrast to previously reported approaches to induce glass formation by increasing the polydispersity of the colloids[18,22], crystallization in our case is prevented while using a single set of monodisperse particles (Supplementary Table 1). Thus, it is likely that the predominantly elastic behavior of the ink is not only important for printing distortion-free three-dimensional structures, but also plays a role in suppressing crystallization of the monodisperse colloidal particles. Indeed, the heat treatment of the structure at 200 °C leads to an estimated linear shrinkage of 14% (Supplementary Note 3) without affecting the long-range disordered arrangement of particles. Structural color emerges from this densification, as demonstrated by photographs taken from a sample treated at different temperatures up to 200 °C (Supplementary Fig. 3).

To quantify the short-range order responsible for the emergence of isotropic structural color, we carried out small-angle X-ray scattering (SAXS) experiments on samples heat treated at 200 °C (Fig. 3d, e). The measured SAXS pattern displays the continuous concentric rings that characterize isotropic structures, confirming the formation of a colloidal glass. Analysis of the azimuthal scattered intensity as a function of the scattering vector (Q) provides more information about the short-range order of the colloidal glass. The intensity peak shown in the azimuthal curve indicates that a specific angle range is selectively scattered by the structure. From the full width at half maximum (ΔQ) of this peak, we estimate the range of spatial order (ξ) leading to such selective scattering using the relation: $\xi = 2\pi/\Delta Q$. For structures formed by 250 nm silica particles, the ΔQ value obtained by fitting a Gaussian distribution to our experimental data leads to a range of spatial order of 699 nm. SAXS measurements on the object prepared from 300 nm particles revealed spatial order of 550 nm for this sample (Supplementary Fig. 4). Both ranges are only few times (2.8 for 250 nm and 1.8 for 300 nm particles) the diameter of the used particles and indicate the short-range order of the formed colloidal glass. The isotropic nature of the structural color arises from the colloidal glass structure of the assembly, in agreement with previous work on similar angle-independent photonic structures[18].

**Emergence, characterization and programmability of color**
The formulation of inks that meet the rheological properties required for DIW and that generate photonic colloidal glasses upon heat treatment enabled the development of a platform for the 3D printing of objects with tunable structural color. To illustrate the potential of such manufacturing platform, we first print three-dimensional grid-like structures, the color of which is tuned by changing the size of the silica colloidal particles present in the initial ink (Fig. 4a–c). The other parameters expected to influence the color of the structure, such as the volume fraction of the particles, the refractive index of the medium and the amount of CB, were kept unchanged in this demonstration except the volume fraction. The volume fraction of the colloids in the

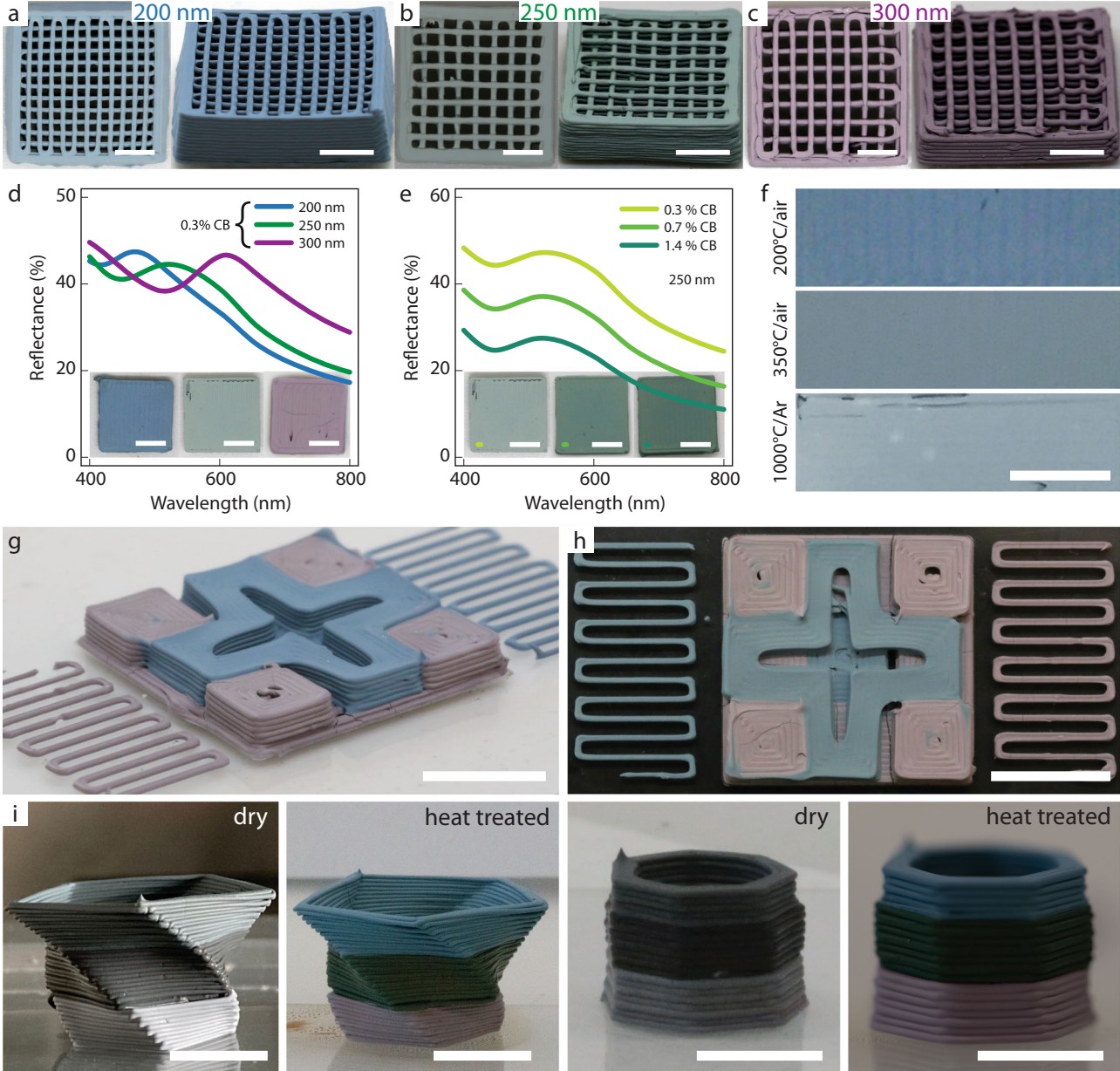

**Fig. 4 | 3D printed objects with isotropic structural color.** Grid-like structures printed from inks containing **a** 200 nm, **b** 250 nm, and **c** 300 nm silica colloids. Reflectance spectra of 3D printed and heat treated specimens prepared using **d** different silica particle sizes and fixed carbon black content of 0.3 wt%, and **e** different carbon black (CB) concentrations and fixed silica particle size of 250 nm. **f** Photographs of printed samples prepared with 200 nm silica particles and 0.3 wt% carbon black after heat treatment at different temperatures and atmospheres. **g, h** Side and top views of a complex object manufactured by multimaterial 3D printing of colloidal inks containing 200 nm (blue) and 300 nm (pink) silica particles. **i** Side views of twisted and hexagonal vases manufactured by multimaterial 3D printing of colloidal inks containing 200 nm (blue), 250 nm (green), and 300 nm (pink) silica particles. Scale bars: 5 mm in (**a**–**e**); 3 mm in (**f**); 1 cm in (**g**–**i**).

ink may slightly vary during ink homogenization (see Supplementary Note 5 for details).

Monodisperse silica particles with diameter of 200, 250, and 300 nm led to distinctive blue, green, and magenta structural colors, respectively. Because the color arises from the nanoscale structure of the material, this approach leads to colored objects that are not limited by the long-term bleaching of conventional pigments and do not require painting after manufacturing. In contrast to the iridescent effects of previously reported photonic structures fabricated by additive manufacturing[38,39,41,43], the color of the printed objects is angle-independent. Reflectance measurements at various angles proves the angle-independent nature of the color (Supplementary Note 6 and Supplementary Fig. 5). Moreover, the shaping capabilities offered by the DIW technique can be fully leveraged, enabling the fabrication of grids with long free-spanning filaments and high local curvature, as predicted from our rheological data (Fig. 2).

To quantify the color generated by the photonic glasses, we measured the reflectance of the printed objects as a function of the wavelength of light for a fixed illumination angle (Fig. 4d). The experimental results indicate the presence of a reflectance peak on the order of 42–45%, which arises from the photonic glass. The wavelength of the reflectance peak consistently moves from 470 to 535 to 610 nm as the diameter of the silica particles changes from 200 to 250 to 300 nm, respectively. Such wavelengths correspond to, respectively, blue, green, and red color.

The optical properties of the printed objects can be analyzed in terms of the scattering behavior of photonic glasses. In contrast to previously reported hollow particles[47], the color of our colloidal glass

emerges from both scattering by a single particle and the interference of scattered waves from the assembly of the particles. These scattering contributions are captured by the form factor and the structure factor of the colloidal system[26,31]. The form factor describes the scattering from particles and can be calculated from Mie theory; the structure factor takes into account the constructive interference of waves scattered by different particles, which can be calculated using the Percus–Yevick equation[48].

We use previously proposed scattering models to explain the reflectance measured for our printed colloidal glasses. The reflectance measured with an integrating sphere is a superposition of singly and multiply scattered light. The scattering differential cross-section (or scattering function) in an isotropic photonic glass displays a distinct structure factor peak in the SAXS measurements at the scattering vector $Q_{peak}$. For dense colloidal glasses $Q_{peak} \sim 2.3 \frac{\pi}{d}$, with $d$ being the diameter of the colloidal particle[49,50]. Taking $d = 250$ nm, we obtain $Q_{peak} \sim 0.0029$ Å$^{-1}$, which is in very good agreement with the experimentally measured value of 0.00285 Å$^{-1}$ (Fig. 3e). To predict the wavelength of the scattered light ($\lambda$) that interfere constructively to generate the reflectance peak, we calculate the wavevector $k$ from the scattering vector $Q_{peak}$ using the relation: $Q_{peak} = 2k \sin\left(\frac{\theta}{2}\right)$, assuming a back-scattering angle $\theta \approx \pi$. For elastic scattering, the wavelength of the scattered light, $\lambda$, can be obtained from the equation: $k = 2\pi \frac{n_{eff}}{\lambda}$, where $n_{eff}$ is the effective refractive index of the medium. Using the Maxwell–Garnett approximation, we estimate the effective refractive index $n_{eff}$ of a medium comprising 60% silica ($n = 1.47$) in air to be 1.28.

From the above analysis, we expect a peak in single scattering back-reflection to occur at $\lambda_{peak} \sim \frac{4}{2.3} n_{eff} d$. For $d = 250$ nm, we obtain a peak wavelength ($\lambda_{peak}$) of 557 nm, which is in good agreement with the peak at 535 nm obtained from the measurements on objects made with 250 nm silica particles (Fig. 4d). When not directly reflected, single scattered light is either lost (absorbed or transmitted) or multiply scattered[27,32]. In our optically thick samples (filament diameter 0.58 mm), we can neglect the transmission pathway. Multiple scattering leads to an undesirable diffuse broadband background which we suppress by adding the broadband absorber carbon black.

The predicted reflectance peak wavelengths (Fig. 4d) are about 2.2 times larger than the monodisperse particles size ($d$), thus providing a simple guideline for the selection of the colloidal particles to be added to the ink depending on the desired structural color. Slight shifts in the peak wavelength can be explained by variations in the solid volume fraction and the relatively broad nature of the spectra, which arises from residual diffusive scattering from the individual particles[27,50].

Reflectance experiments were also conducted to evaluate the effect of carbon black as broad-band absorber on the photonic properties of the colloidal glass (Fig. 4e). For a fixed silica particle size, an increase of the carbon black concentration from 0.3 to 1.4 wt% was found to decrease the reflectance of the object at the green peak $\lambda = 550$ nm from 47 to 28% without shifting the position and the width of the reflectance peak. This observation is in line with the light absorption properties of the carbon black particles, which reduce the amount of light reflected by the surface. Importantly, the absorption of photons by the carbon particles ensures that only the light that is strongly scattered by particles located at a small distance from the surface contributes to the overall reflection of the object.

This is due to the fact that CB reduces the amount of multiple scattering. To minimize multiple scattering, the CB absorption or extinction path length has to be comparable to or slightly larger than the mean-free path. The mean free path is the length light can travel without being scattered. The goal of adding CB is to allow only single scattering events to take place, by ensuring that the light that penetrates the structure by multiple scattering is rapidly absorbed. According to previous modeling work[31], the effective mean free path of carbon-containing structures similar to our system varies from 1 to 10

µm over the wavelength range studied. The constant width of the reflectance peaks suggests that the carbon black concentration range used in our inks is sufficient to suppress multiple scattering events that would otherwise add a wavelength-independent scattering background, leading to a broad peak and white color[18,31] (see Supplementary Fig. 6).

A particular feature of our structurally colored printed objects is their thermal stability. Because the colloidal glass is made out of silica particles, exposing them to high temperatures does not change significantly the color reflected by the object (Fig. 4f). Indeed, samples containing 200 nm silica particles retain their original blue color after subjected to a heat treatment at 350 °C for 1 h in air. The color intensity is slightly reduced, probably due to the partial oxidation and removal of the carbon black. This decreases broad-band absorption by the continuous phase, resulting in a slight whitening effect similar to that observed for samples prepared with lower carbon black concentrations. By using a protective argon atmosphere, we found that the blue color can be preserved even after heating the object up to 1000 °C (Fig. 4f). Reflection spectra measurements on samples treated at 200 and 1000 °C show that the higher temperature increases the overall broadband reflectance associated with multiple scattering without affecting the spectral pattern (see Supplementary Fig. 7). Such high heat resistance allows the colored objects to be processed and applied over a broad temperature window, leading to a clear advantage compared to temperature-sensitive pigments. Although we have used a nozzle diameter of 410 µm in this work, increasing the resolution of the print lines can in principle enhance the saturation of the color or decrease the level of CB needed for a similar color, as previously suggested[51].

To fully exploit the potential of our 3D printing platform, we manufactured a centimeter-scale object with dual and ternary structural colors using the spatial control and multimaterial capabilities offered by the DIW technique (Fig. 4g, i). The objects combine specific shapes and colors into a three-dimensional pattern that resembles the Swiss flag or twisted and hexagonal vases (Fig. 4i). To enable multimaterial printing of such an object, inks containing monodisperse silica particles with diameters of 200 nm, 250 nm, or 300 nm are loaded in separate cartridges of the printer. Printing was carried out through the sequential deposition of the two inks using a computer-generated design. As-printed structures were heat treated at 200 °C to extract the aqueous phase of the ink and generate photonic glasses with the pre-programmed structural colors. The final heat-treated object displays vivid blue and magenta colors and high fidelity to the original design without any color diffusion or distortion (see Supplementary Fig. 8). Most importantly, the structural nature of the color allows for the fabrication of a multi-colored object using predominantly the same chemical composition throughout the bulk and surface of the material. This resembles the strategy used by living organisms to generate a wide range of functional properties using a limited selection of abundant and sustainable chemistries[52–54]. With a silica weight fraction of over 97%, our objects are recyclable and do not display the toxicity issues of typical colored pigments. As opposed to paintings and coatings, chipping of the surface does not compromise its optical functionality, since the color is an intrinsic property of the bulk material. Easy recycling can be achieved via mechanical pulverization of objects and further addition of CB and PEO-PPO-PEO copolymer to yield a new ink. Moreover, the changes in color expected by infiltrating the colloidal glass with different liquids can potentially be explored to imbue the object with attractive sensing capabilities.

Complex-shaped objects with isotropic structural color can be manufactured by 3D printing aqueous inks containing monodisperse silica particles. Structural color results from selective scattering from the colloidal glass formed upon printing and heat treatment. The manufacturing of the structurally colored objects relies on the formulation of inks displaying the rheological properties required for

DIW and leading to the formation of a dense colloidal glass with periodicities at length scales comparable to visible wavelengths. These criteria can be satisfied by formulating inks with a volume fraction of particles that is sufficiently low to enable flow through the printing nozzle ($\phi$ = 0.32), but also high enough to form the desired colloidal glass upon complete removal of water from the printed object. The peak wavelength of the color generated by the photonic glass was found to be on the order of the size of the particle and could therefore be easily tuned by varying the diameter of the particles used. Besides the selectively scattering particles, vivid structural color is only achieved by incorporating carbon black in the ink to provide a broadband absorbing medium within the colloidal glass. A gel-forming copolymer is also present in the ink to reach the viscoelastic properties needed for printing. Our approach provides a powerful tool to design and fabricate multi-colored 3D objects through the multimaterial printing of inks of programmable structural color. Just like photonic materials created by living organisms in nature, our structurally colored objects can be produced using widely available, non-toxic, and sustainable chemistries.

## Methods

### Preparation of colloidal inks

For the preparation of a typical ink, a 0.84 mL aliquot of concentrated ammonia solution (Ammonia solution 25% for analysis, Emsure), 0.04 g of polycarboxylate dispersant (Dolapix CE 64, Schimmer & Schwarz, Germany), and an aliquot of carbon black suspension (CB, Cabot VXC72R, provided by BCD-Chemistry, Germany) were added into a 20 g solution of 28.6 wt% PEO-PPO-PEO (Pluronic F-108, Sigma Aldrich). The aqueous CB suspension contained 10% w/w CB and 2% w/w PEO-PPO-PEO (Pluronic F-108, Sigma Aldrich). The amount of CB suspension added to the ink was tuned so as to achieve a final CB content of 0.3, 0.7, and 1.4 wt% in the ink. 20 g monodisperse silica nanoparticles were added in powder form to the aqueous mixture of carbon black, ammonia, dispersant and PPO-PEO-PPO. Silica particles of 200, 250, and 300 nm were purchased from Fiber Optic Center Inc. (AngstromSphere Monodisperse Silica Powder). The monodispersity and size range of the particles were measured by Dynamic Light Scattering (Table S1). The final ink was homogenized with a planetary mixer (ARE-250, Thinky) at 2000 rpm for two rounds of 1 min with two ceramic balls in the container. Between the two mixing steps, the ink was cooled in an ice bath for 10 min to decrease the viscosity of the PEO-PPO-PEO solution. The ink was processed three times on a three roll-mill (DSY-200, Bühler, Switzerland). During this step, the gap size between the rolls was progressively decreased from 120 to 5 μm. In a typical ink formulation, the fraction of silica particles was fixed at 32 vol%. The final mixed ink showed a homogeneous dark color and an even distribution of CB within the silica colloids (Supplementary Figs. 9 and 10).

### Three-dimensional printing

The colloidal inks were loaded into syringes and centrifuged at 2000 rpm for 1 min to remove bubbles prior to printing. Syringes loaded with the inks were mounted in a commercial extrusion-based 3D printer (3D Discovery, RegenHU Ltd., Switzerland). During printing, the ink was pneumatically extruded through a micronozzle onto hydrophobized glass substrates under an applied pressure of 2–3 bars. Nozzle diameters of 410 μm or 580 μm were used to achieve reasonable resolution, while preventing clogging during extrusion (Supplementary Note 7). After printing, the materials were dried under a cover at ambient temperature. Samples were also heat treated at 200 °C for 1 h for complete removal of water. During this heating step, the temperature was increased slowly using a ramping period of 3 h to minimize cracking. Note that all the printed structures shown in this work include carbon black.

### Rheology

Rheological characterization of the inks was performed using a plate-plate geometry mounted on a strain- or stress-controlled rheometer (MCR501 and MCR702, Anton Paar). Oscillatory amplitude sweeps were carried out to measure the storage ($G'$) and loss ($G'$) moduli of the ink. Tests were conducted at a frequency of 10 rad/s by increasing the shear strain amplitude from 0.1 to 100%. The apparent viscosity of the inks was quantified through steady-shear measurements conducted by progressively increasing the applied shear rate from 0.1 to 100 s$^{-1}$. To emulate the printing process and measure the elastic recovery of the ink, samples were first subjected to an oscillatory time sweep with 1% strain and 1 rad s$^{-1}$, followed by a steady-shear period of 1 min at 10 s$^{-1}$ before applying another oscillatory time sweep under the same conditions used at the beginning of the test (Fig. 3c). Finally, the yield stress of the ink was quantified by applying an increasing shear stress and measuring the resulting shear strain.

### Microstructural characterization

The structure of printed and dried samples was characterized using scanning electron microscopy (SEM, LEO 1530 Gemini microscope). The SEM was equipped with an in-lens detector and was operated at an acceleration voltage of 2 kV, an aperture size of 30 μm, and a working distance of 3–5 mm.

### Optical properties

The diffuse reflectance of printed and heat-treated samples was measured with spectral resolution by using an integrating sphere and a spectrometer for visible wavelengths (Ocean Optics Inc.).

### Small-angle X-ray scattering

The scattering pattern of printed colloidal structures was characterized for samples with silica particle diameters of 250 and 300 nm (Fig. 3d, e and Supplementary Fig. 2). SAXS/WAXS X-ray scattering measurements were performed with a Xeuss 3.0 UHR SAXS/WAXS system (Xenocs SAS, Grenoble, France) equipped with a GeniX 3D Cu Kalpha radiation source ($\lambda \approx 1.54$ Å) and a Q-Xoom in-vacuum motorized Eiger2 R 1M detector (Dectris Ltd., Switzerland) for data collection. 2D data reduction into 1D plots of scattering intensities, $I$(q), as a function of momentum transfer, $Q$ ($Q = 4\pi \sin \theta/\lambda$, $\theta$ is half the scattering angle) was performed automatically using the Xenocs XSACT software.

### Thermogravimetric analysis

Thermogravimetric analysis (TGA) was performed using a Mettler Toledo TGA / DSC 3+ system in the range from 25 to 800 °C in air using a ramp of 10 °C min$^{-1}$.

## Data availability

The data supporting the findings of this study are available from the corresponding authors upon request. In addition, the SAXS, rheology, reflection, light scattering, and TGA data generated in this study have been provided as a Supplementary Data to the paper. Source data are provided with this paper.

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

## Acknowledgements

We gratefully acknowledge Xenocs France for provision of beam time on the Xeuss USAXS/SAXS/WAXS UHR equipment and Dr. P. Panine for the related SAXS/WAXS experimental work and the calculation of the ΔQ value. We also thank Vladimir Vojtech, Prof. Jörg Löffler, Dr. Tian Liu, Prof. Markus Niederberger, Prof. Ralph Spolenak and Prof. Eric Dufresne, ETH Zurich for the access to the instrument. This research was supported by ETH Zürich and the Swiss National Science Foundation through the National Center of Competence in Research Bio-Inspired Materials (grant number: 51NF40_182881) and through a personal grant (grant number 188484). This research was in part supported by São Paulo Research Foundation - FAPESP (grant #2015/20052-0, #2018/06456-0 and #2020/02938-0).

## Author contributions

A.F.D. and E.P. conceived and initiated the project, designed the experiments. M.C., E.P., and A.F.D. prepared the formulations and performed the experiments, and analyzed the data. M.R.B., F.L.B., F.B.C., L.D.C.d.C., W.W., and N.K. helped with 3D printing, CAD design, characterization, and analyses. A.S., H.G., and F.S. performed and conceived the optical characterization and analyses. A.F.D., E.P., and A.R.S. supervised the research and wrote bulk of the paper. All authors revised and edited the paper.

## Competing interests

The authors declare no competing interests.
