## [Peer Review File · Nature Communications]

Three-dimensional Printing of Photonic Colloidal Glasses into Objects with Isotropic Structural ColorREVIEWER COMMENTS

Reviewer #1 (Remarks to the Author):

In the manuscript, the authors report the formulation of a water-based colloidal ink and the 3D printing with it into objects with isotropic structural colours. The colours are produced due to the formation of photonic glasses via the self-assembly of the silica nanoparticle in the ink. Through the control of several ink parameters, they managed to fabricate different shapes of objects with different colours. While the work is interesting; however, it is too preliminary to be published at this stage. If the authors can address the following questions, it can be further considered for publication.

Major questions:

1. I don't think that I will agree with the statement the authors made at the end of the third paragraph: "However, current additive manufacturing methods have been restricted to crystalline structures with angle-dependent photonic properties, preventing a broader application of structural colour as an alternative to conventional pigments and dyes." In the paper <https://www.science.org/doi/epdf/10.1126/sciadv.abj8780>, they have already reported the direct writing of crystalline structures as well as glasses structures for structural-colour design. This previous work, therefore, reduces a bit the originality of the presented manuscript.
2. I do not understand the mechanism reported in Figure 3b, the weight loss increases at 150°C where there is a small peak. Can the author better clarify this point?
3. The monodispersity or polydispersity of the silica nanoparticles has not been evaluated. The authors should evaluate the polydispersity of the purchased nanoparticles Fiber Optic Center Inc. (AngstromSphere Monodisperse Silica Powder). The polydispersity of the nanoparticles will have a great influence on the self-assembly of these nanoparticles, to properly claim that they have a photonic glass the Mie response looks weak. Also, from the SEM image (Figure 3c), the nanoparticles span quite a large range in size.
4. In page #7, below Figure 3, it is not very precise to state "the color of which is determined by the size of the silica colloidal particles present in the initial ink (Figure 4a-c)." It is true that the size of the silica nanoparticles can affect the colours. However, what determines the colours, in this case, is the structure factor, (i.e. how the scattered light from nanoparticles is coherent with each other), which at the end is related to the averaged centre to centre distance between nanoparticles and the refractive indexes of nanoparticles and the medium. The authors need to add more discussion somewhere to better explain this point
5. Following the last point, can the authors also discuss the spectra not only relative to the nanoparticle size and have a more comprehensive understanding of the physics of structural colours from photonic glasses? For example, for Figure 4d.
6. For Figure 4e, it is not true that an increase in the carbon black concentration decreases the reflectance of the object. From 0 to 0.3wt%, comparing the spectra in Figures 4d and 4e, the reflectance increases. This point needs to be addressed.
7. I do not see any points to blend carbon black in this system. It is supposed to be used to improve the colour purity in some systems by absorbing randomly scattered light and narrowing the reflection peak. However, it does not seem to improve the response and it

actually decreases the intensity.

8. In page #8, the authors state “Importantly, the absorption of photons by the carbon particles ensures that only the light that is strongly scattered by particles located at a small distance from the surface contribute to the overall reflection of the object.” Could the authors specify how small the distance needs to be?

9. I don't think I would agree with the statement:” The constant width of the reflectance peaks suggests that the carbon black concentration range used in our inks is sufficient to suppress multiple scattering events that would otherwise lead to a broad peak and white color 18, 31.” The spectra of samples with carbon black are the same in reflection peak width as those without carbon black. How can the authors draw this conclusion from these spectra?

10. In Figure 4f, the authors believed that “By using a protective argon atmosphere, we found that the blue colour can be preserved even after heating the object up to 1000 °C, see Figure 4f. Such high heat resistance allows the coloured objects to be processed and applied over a broad temperature window, leading to a clear advantage compared to temperature-sensitive pigments.” Could the authors compare the optical properties of these samples more scientifically, such as by spectroscopy? From the picture reported it actually looks like the the colour has changed after heating.

11. For Figure 4g and 4h, what is the best resolution for the printing they can get? How can they improve that?

12. The authors claimed they can print complex-shaped objects? Can the authors provide more complex printed objects than the Swiss flag? Could they do a 3D Eiffel tower with the printing ink and the printing setup they have now?

Minor points:

1. The authors need to standardise the references. For example, in 32, 35, and 22, you put the citation outside the full stop.

2. How can the authors recycle the objects?

Reviewer #2 (Remarks to the Author):

3D printed patterns of colloidal glasses were experimentally demonstrated for isotropic structural colors, which is never reported. Figures are well organized and manuscript is well written. Therefore, I would like to recommend publication after addressing minor issues.

- Since this work intended to show the potential of photonic glasses in practical application, authors should demonstrate the various lines of photonic glasses with different diameter and their reluctance spectra should be compared. Authors should discuss whether the diameter of line affect the reflectance spectra. What would be minimum size of line for structural coloration?

- Also, in single line on substrate, photonic glasses in line may be dried anisotropically. In other words, only height may be reduced during the drying process, which may affect the angle-dependent color of photonic glasses. Therefore, cross-sectional SEM or TEM images of photonic glasses should be added after drying and structure of silica particles should be analyzed.
- Angle-dependent colors of photonic glass line should be confirmed in reflectance spectra.
- It would be very helpful to show how structural color is developed as a function of the drying time.
- In line 51, angle-independent scattering should be corrected as 'angle-independent multiple scattering'.
- In Figure S1, many micron-sized holes are observed. Authors should explain what are those and what caused that.
- Is there any mixing between lines with different photonic glasses during drying process?
- Authors should mention the isotropic structural colors from form factor in introduction including references (Small 15(23), 1900931 (2019) and others).

RESPONSE TO REFEREE COMMENTS**Reviewer comments:****Reviewer #1 (Remarks to the Author):**

In the manuscript, the authors report the formulation of a water-based colloidal ink and the 3D printing with it into objects with isotropic structural colours. The colours are produced due to the formation of photonic glasses via the self-assembly of the silica nanoparticle in the ink. Through the control of several ink parameters, they managed to fabricate different shapes of objects with different colours. While the work is interesting; however, it is too preliminary to be published at this stage. If the authors can address the following questions, it can be further considered for publication.

General reply: We thank the Reviewer for thoroughly reading our paper and for providing detailed and helpful comments. We are also glad to hear that she/he agrees with the main conclusions of our work and finds our work interesting. Below, we address all remarks of the reviewer.

Major questions:

1-I don't think that I will agree with the statement the authors made at the end of the third paragraph: "However, current additive manufacturing methods have been restricted to crystalline structures with angle-dependent photonic properties, preventing a broader application of structural colour as an alternative to conventional pigments and dyes." In the paper <https://www.science.org/doi/epdf/10.1126/sciadv.abj8780>, they have already reported the direct writing of crystalline structures as well as glasses structures for structural-colour design. This previous work, therefore, reduces a bit the originality of the presented manuscript.

Reply: We thank the reviewer for this valuable comment and bringing this work to our attention, which we cite now in the revised manuscript. Although this work is demonstrating examples of both photonic crystals and glasses, we believe that most of the demonstrations in the paper are limited to complex 2D structures printed on a substrate. In contrary, we have shown bulk 3D objects with angle-independent and continuous coloration. In the revised version we have also increased the complexity of the structures we have printed by adding twisted vase and hexagonal vase structures with multiple colors to the library of possible 3D objects. We also note that these possibilities are meant to showcase the capabilities of the platform. We now cite this relevant work and add the following text:

The direct-writing approach has also been applied to self-assemble monodisperse particles into colorful colloidal crystals³⁹ and disordered structures⁴⁴. However, current additive manufacturing methods have been restricted either to 2D demonstrations⁴⁴ or to 3D crystalline structures with angle-dependent photonic properties³⁹, preventing a broader application of structural color as alternative to conventional pigments and dyes.

2. I do not understand the mechanism reported in Figure 3b, the weight loss increases at 150°C where there is a small peak. Can the author better clarify this point?

Reply: We thank the reviewer for this comment. The plot in Figure 3b (above) shows the thermal gravimetric analysis of a 3D printed green body. The left-hand side of the plot shows the weight loss in % and the right-hand side shows the Heat flow to the sample. We added the colors and arrows to make them clearly distinguishable.

To further understand and interpret the different parts of the TGA weight loss curve, we have now performed TGA analyses of the individual components of the 3D printed sample. We namely conducted TGA of carbon black (CB), PEO-PPO-PEO copolymer (Pluronic F108) and the silica particles, in addition to the analyses of the 3D printed green body. The new measurements are shown below. We clearly observe that the CB does not lose significant weight until 700 °C. Silica loses 5% of its weight until 400 °C. Interestingly, the PEO-PPO-PEO copolymer exhibits a strong weight loss at 300°C and loses almost all of its weight at 400 °C. This is similar to what we observe in the curve of the 3D printed green body, but at a higher temperature. In the light of these analyses, we believe that the initial loss up to 200°C is due to the removal of the remaining water from the printed object. The small peak at 200°C is likely due to uptake/absorbance of oxygen on the dried materials as a result of oxidation. It is likely the oxidation of the copolymer, because this molecule starts degrading at this temperature. The weight loss above 200°C is due to the degradation of the copolymer.

The weight percent of this loss corresponds well with the weight % of the copolymer added in the ink. Upon removal of the copolymer, the particle assembly densifies and the structural color emerges. The TGA analysis shown in Figure S1(f) simulates our heating cycle and indicates a ~90% weight loss of the copolymer alone after a heat treatment of 1 h at 200°C. This is also clear evidence that the copolymer decomposes during the heat treatment. Here, we also observe that the decomposition temperature of the pure copolymer is 300°C. This temperature shifts to 200°C in the ink mixture. Such shift is probably related to the fact that the copolymer is hydrophilic and it swells in an aqueous solution at ambient temperature, making it more prone to degradation. When mixed with silica and carbon black, the copolymer might also adhere or adsorb on the surface of these particles, which affects its stability and decreases the decomposition temperature. To address this new understanding of the system, we made the following changes to the main text and added the Supplementary Figure S1 and accompanying discussion. We have also replaced throughout the text the word 'drying' by 'heat treatment' for sake of clarity.

Supplementary Figure S1. Thermal gravimetric analysis (TGA) of the ink components. TGA curves taken from 30 to 800°C of (a) PEO-PPO-PEO copolymer (Pluronic F108), (b) 250-nm silica particles, (c) carbon black and (d) the produced ink. (e) Combined plot of curves shown in (a-d). (f) TGA of the PEO-PPO-PEO copolymer alone under isothermal conditions at 200°C. Here, the sample was heated at a speed of 10°/min up to 200 °C and the temperature was held at 200°C for 1 h to simulate our drying experiments.

Thermal gravimetrical analysis of the ink and the ink constituents

To understand and interpret the different parts of the TGA curve shown in the main text (Figure 3b), we have also measured the thermally induced weight loss of the individual constituents of the 3D printing ink. TGA was conducted for the PEO-PPO-PEO copolymer (Pluronic F108), silica particles, carbon black (CB) and for the 3D printed green body after room-temperature drying (Figure S1). As expected, the silica and CB particles show only relatively low mass losses under the investigated temperature range. The 5% weight loss observed for the silica particles until 400 °C is likely due to the evaporation of physically and chemically bound water or possibly the decomposition of organics added during synthesis (Figure S1b). CB starts to lose weight only above 700 °C, when carbon is expected to be oxidized (Figure S1c).

Interestingly, the PEO-PPO-PEO copolymer exhibits a strong weight drop at 300°C and loses almost all its mass at 400 °C (Figure S1a). By comparing the TGA curve obtained for the whole ink with the measurements of the individual components (Figure S1e), we find that the ink weight loss at 200°C is probably due to the decomposition of the PEO-PPO-PEO copolymer at this temperature. Indeed, the weight percentage of this loss corresponds well with the weight percentage of the copolymer in the ink. Upon removal of the copolymer, the particle assembly densifies and the structural color emerges.

The reduction in decomposition temperature from 300°C to 200°C observed when the copolymer is incorporated into the ink is probably related to the different configuration of the molecules in the aqueous suspension. The PEO-PPO-PEO copolymer is a hydrophilic molecule that swells in an aqueous solution at ambient temperature. Moreover, these molecules are expected to adsorb on the surface of the silica and carbon black particles present in the ink. These factors likely reduce the thermal stability of the copolymer, thus lowering its decomposition temperature.

To gain further insights into the thermal decomposition of the copolymer alone, an additional TGA measurement was performed under the same heat treatment applied to the printed objects (Figure S1f). In this experiment, the temperature is held at 200°C after an initial quick heating of the sample. The results show that the copolymer loses ~90% of its weight after a 1-hour heat treatment at 200°C. This confirms that the emergence of color during heat treatment results from the densification that takes place upon the removal of water and copolymer from the printed assembly.

3. The monodispersity or polydispersity of the silica nanoparticles has not been evaluated. The authors should evaluate the polydispersity of the purchased nanoparticles Fiber Optic Center Inc. (AngstromSphere Monodisperse Silica Powder). The polydispersity of the nanoparticles will have a great influence on the self-assembly of these nanoparticles, to properly claim that they have a photonic glass the Mie response looks weak.

Also, from the SEM image (Figure 3c), the nanoparticles span quite a large range in size.

Reply: We agree with the Reviewer that the polydispersity of the particles influence the self-assembly process significantly. However, we also would like to note here that increasing polydispersity or use of bi-disperse particles have been used as strategies to hinder crystallization and in order to promote glass formation (see ref doi.org/10.1002/cphc.200900869 and [10.1002/adma.200903693](https://doi.org/10.1002/adma.200903693)). Note that the high polydispersity increases the likelihood of particle assembly into glasses. Indeed, the SAXS measurements revealed no bright diffraction spots, but ring patterns that clearly demonstrate the amorphous state of the assembly, which we refer to as a photonic glass. Together with the newly added angle-independent reflection spectra (Supplementary Figure S5) and the SAXS measurements (Figure 3d, 3e) we believe it is reasonable to call our system photonic glasses. Following the reviewer's suggestion, we have now analysed the commercial silica particles in terms of size and polydispersities (Table S1). The measured polydispersity values indicate that the particles are highly monodisperse. Note that the polydispersities of the particles are reported by the company to be <10%.

Supporting Table

Table S1. Average hydrodynamic diameter and polydispersity of the silica particles measured by Dynamic Light Scattering (DLS). *

AngstromSphere reported size [nm]	Light Scattering size [nm]	Polydispersity [%]
200	234	2
250	256	2
300	311	6

* The particle size analyses (DLS) were performed with a Malvern Zetasizer Nano ZS instrument. The power of DLS to quantify small polydispersities is limited, but for a measured DLS-polydispersity << 10% it sets an upper bound of approximately 10% in agreement with the supplier specification⁵⁵.

4. In page #7, below Figure 3, it is not very precise to state “the color of which is determined by the size of the silica colloidal particles present in the initial ink (Figure 4a-c).” It is true that the size of the silica nanoparticles can affect the colours. However, what determines the colours, in this case, is the structure factor, (i.e. how the scattered light from nanoparticles is coherent with each other), which at the end is related to the averaged centre to centre distance between nanoparticles and the refractive indexes of nanoparticles and the medium. The authors need to add more discussion somewhere to better explain this point.

Reply: In accordance with the agreeable suggestion of the Reviewer we now replaced our text with the following in our main text. See also the response to the next comment.

To illustrate the potential of such manufacturing platform, we first print three-dimensional grid-like structures, the color of which is tuned by changing the size of the silica colloidal particles present in the initial ink (Figure 4a-c). The other parameters expected to influence the color of the structure, such as the volume fraction of the particles, the refractive index of the medium and the amount of CB, were kept unchanged in this demonstration.

5. Following the last point, can the authors also discuss the spectra not only relative to the nanoparticle size and have a more comprehensive understanding of the physics of structural colours from photonic glasses? For example, for Figure 4d.

Reply: In accordance with the suggestion of the review we now replaced our original text with the following discussion on page 8.

The optical properties of the printed objects can be analyzed in terms of the scattering behavior of photonic glasses. In contrast to previously reported hollow particles⁴⁷, the color of our colloidal glass emerges from both scattering by a single particle and the interference of scattered waves from the assembly of the particles. These scattering contributions are captured by the form factor and the structure factor of the colloidal system^{26,31}. The form factor describes the scattering from individual particles and can be calculated from Mie theory; the structure factor takes into account the constructive interference of waves scattered by different particles, which can be calculated using the Percus–Yevick equation⁴⁸.

We use previously proposed scattering models to explain the reflectance measured for our printed colloidal glasses. The reflectance measured with an integrating sphere is a superposition of singly and multiply scattered light. The scattering differential cross-section (or scattering function) in an isotropic photonic glass displays a distinct structure factor peak in the SAXS measurements at the scattering vector Q_{peak} . For dense colloidal glasses $Q_{peak} = 2.3 \frac{\pi}{d}$, with d being the diameter of the colloidal particle.^{49,50} Taking $d = 250\text{nm}$, we obtain $Q_{peak} = 0.0029 \text{ \AA}^{-1}$, which is in very good agreement with the experimentally measured value of $0,00295 \text{ \AA}^{-1}$ (Figure 3e). To predict the wavelength of the scattered light (λ) that interfere constructively to generate the reflectance peak, we calculate the wavevector k from the scattering vector Q_{peak} using the relation: $Q_{peak} = 2k \sin\left(\frac{\theta}{2}\right)$, assuming a back-scattering angle $\theta \approx \pi$. For elastic scattering, the wavelength of the scattered light, λ , can be obtained from the equation: $k = 2\pi \frac{n_{eff}}{\lambda}$, where n_{eff} is the effective refractive index of the medium. Using the Maxwell-Garnett approximation, we estimate the effective refractive index n_{eff} of a medium comprising 65% silica ($n = 1.47$) in air to be 1.3.

From the above analysis, we expect a peak in single scattering back-reflection at 563 nm, which is in good agreement with the peak at 535 nm obtained from the measurements on objects made with 250 nm silica particles (Figure 4d). When not directly reflected, single scattered light is either lost (absorbed or transmitted) or multiply scattered^{27,32}. In our optically thick samples (filament diameter 0.58mm), we

can neglect the transmission pathway. Multiple scattering leads to an undesirable diffuse broadband background which we suppress by adding the broadband absorber carbon black.

6. For Figure 4e, it is not true that an increase in the carbon black concentration decreases the reflectance of the object. From 0 to 0.3wt%, comparing the spectra in Figures 4d and 4e, the reflectance increases. This point needs to be addressed.

Reply: We believe there is a misunderstanding here. We would like to draw the attention of the Reviewer to the fact that the spectra in Fig 4d also include CB. These samples shown in Fig 4d have **0.3% CB** content. In addition, please note that the scale of the plots in Fig. 4 d and 4e are different. To clarify this misunderstanding, we now inserted the CB amount of the samples shown in Fig. 4d. We show the updated Figure 4d below.

7. I do not see any points to blend carbon black in this system. It is supposed to be used to improve the colour purity in some systems by absorbing randomly scattered light and narrowing the reflection peak. However, it does not seem to improve the response and it actually decreases the intensity.

Reply: We are sorry for this unclarity. The color of the 3D printed objects without the presence of CB is dominated by the multiple scattering and therefore looks milk-white. Our samples are optically extremely dense (filament diameter 0.58 mm); therefore, in the absence of an absorber, all light will be diffusely reflected at any relevant wavelength. To obtain a saturated color we always needed to add CB to our samples, so as to reduce diffuse scattering. All samples demonstrated in this work contain CB. To clarify this point, we have now inserted in the legend of the Fig 4d the CB amount of the samples measured (see the response to the above comment). We also added the following statement in the Material methods section.

Note that, all the printed structures shown in this work include carbon black.

8. In page #8, the authors state “Importantly, the absorption of photons by the carbon particles ensures that only the light that is strongly scattered by particles located at a small distance from the surface contributes to the overall reflection of the object.” Could the authors specify how small the distance needs to be?

Reply: We thank the reviewer for raising this point. Here what we refer as the ‘distance from the surface’ is the mean-free path of the light. Because CB suppresses multiple scattering events, we assume the measured reflectance to come from single scattering events only. Under this assumption, the light should penetrate the structure with a depth comparable to the photon mean-free path length and has to scatter back from the structure. This mean-free path of light correlates to the CB concentration, as previously described by Schertel et al (Ref. 31). Since this has been previously described in the literature, we did not repeat the analysis here. The effective photon mean-free path of the system decreases with the CB concentration and lies between 1 - 10 microns in our samples. To address this, we added the following statement:

This is due to the fact that CB reduces the amount of multiple scattering. To minimize multiple scattering, the CB absorption or extinction path length has to be comparable to or slightly larger than the mean-free path. The mean free path is the length light can travel without being scattered. The goal of adding CB is to allow only single scattering events to take place, by ensuring that the light that penetrates the structure by multiple scattering is rapidly absorbed. According to previous modelling work³¹, the effective mean free path of carbon-containing structures similar to our system varies from 1 to 10 μm over the wavelength range studied.

9. I don’t think I would agree with the statement:” The constant width of the reflectance peaks suggests that the carbon black concentration range used in our inks is sufficient to suppress multiple scattering events that would otherwise lead to a broad peak and white color 18, 31.” The spectra of samples with carbon black are the same in reflection peak width as those without carbon black. How can the authors draw this conclusion from these spectra?

Reply: As discussed before all our samples contain carbon black. Still, we thank the reviewer for careful reading. We modified the sentence to clarify this point and explain that multiple scattering would add a wavelength-independent scattering background to the system and result in a reflection peak that is shallower and less pronounced. What is meant here is that our inks have this necessary CB amount to avoid this unwanted effect.

The constant width of the reflectance peaks suggests that the carbon black concentration range used in our inks is sufficient to suppress multiple scattering events that would otherwise add a wavelength-independent scattering background, leading to a broad peak and white color^{18,31}.

10. In Figure 4f, the authors believed that “By using a protective argon atmosphere, we found that the blue colour can be preserved even after heating the object up to 1000 °C, see Figure 4f. Such high heat resistance allows the coloured objects to be processed and applied over a broad temperature window, leading to a clear advantage compared to temperature-sensitive pigments.” Could the authors compare the optical properties of these samples more scientifically, such as by spectroscopy? From the picture reported it actually looks like the the colour has changed after heating.

Reply: It is true that the color of the 1000°C sintered sample is a little pale compared to the one dried at 200°C. We now performed light scattering experiments of these samples and report it below as the new supplementary Figure S6 of our manuscript. We performed two set of experiments: one with a near incidence reflection setup and another with the integration sphere (similar to Figure 4d). The reflectance plots for the two samples exhibit a similar profile but differ in their overall reflectance. This we attribute to a partial degradation of the CB due to oxygen (or oxides) present in the sample. As reported previously, lower CB amounts reduces the absorption and thus increases broadband diffuse reflection (see also Figure 4e). We now inserted the following text to the paper where we refer to this new supplementary Figure.

By using a protective argon atmosphere, we found that the blue color can be preserved even after heating the object up to 1000 °C (Figure 4f). Reflection spectra measurements on samples treated at 200 °C and 1000°C show that the higher temperature increases the overall broadband reflectance associated with multiple scattering without affecting the spectral pattern.

Figure S6. Reflectance measurements of samples prepared with 200 nm silica particles dried at 200°C and 1000°C. (a) Near incidence angle reflectance of the samples measured up to 3 times. (b) Total reflectance of the two samples performed with an integrating sphere. (c) Colors calculated from the total reflectance plots in (b) are compared with the photographs of the samples.

11. For Figure 4g and 4h, what is the best resolution for the printing they can get? How can they improve that?

Reply: For DIW, the lower limit of resolution is usually around 100 μm , but can be extended down to a few microns by printing in a liquid at high extrusion pressures (<https://onlinelibrary.wiley.com/doi/10.1002/adma.200602906>). Here, we used a 410 μm nozzle during the printing process and we report this in the 'Materials& methods' section. DIW of particle-containing inks below 100 μm resolution becomes challenging as the possibility of clogging of the nozzle gets higher below this size. This originates from the fact that our ink has a high concentration of particles. Therefore, any particle aggregates in the suspension have to be disintegrated within the ink. We perform this by 3 roll milling the suspension, which reproducibly gives us homogenous inks that can be flawlessly ejected from 410 μm nozzles. In principle, ball-milling dispersions of nanoparticles over long hours may help to further increase the homogeneity of the ink, but this was out of the focus of the current work.

12. The authors claimed they can print complex-shaped objects? Can the authors provide more complex printed objects than the Swiss flag? Could they do a 3D Eiffel tower with the printing ink and the printing setup they have now?

Reply: We thank the reviewer for raising this point. Indeed, the 3D printing method we use allows for nearly limitless 3D shaping possibilities. In addition to complex shapes, the multi-material capabilities of this method allows us to perform multi-color printing and to introduce decorative and functional patterns on top of intricate objects. Following the reviewer's suggestions, we now added a few more examples of complex 3D structures and multicolor objects to our portfolio to demonstrate the versatility of the method in complexity and shape freedom. The new demos of twisted multicolor vase and a multicolor hexagonal vase are added to the Figure 4, which we display below.

Figure 4: 3D printed objects with isotropic structural color. (a-c) Grid-like structures printed from inks containing (a) 200 nm, (b) 250 nm, and (c) 300 nm silica colloids. (d,e) Reflectance spectra of 3D printed and **heat treated** specimens prepared using (d) different silica particle sizes and fixed carbon black content of 0.3 wt%, and (e) different carbon black concentrations and fixed silica particle size of 250 nm. (f) Photographs of printed samples prepared with 200 nm silica particles and 0.3 wt% carbon black after heat treatment at different temperatures and atmospheres. (g-h) Side and top views of a complex object manufactured by multimaterial 3D printing of colloidal inks containing 200 nm (blue) and 300 nm (pink) silica particles. **(i) Sideviews of a twisted and hexagonal vase object manufactured by multimaterial 3D printing of colloidal inks containing 200 nm (blue), 250 nm (green), and 300 nm (pink) silica particles.** Scale bars: 5 mm in (a-e); 3 mm in (f); 1 cm in (g-i).

Minor points:

1. The authors need to standardise the references. For example, in 32, 35, and 22, you put the citation outside the full stop.

Reply: We thank the reviewer for pointing out this mistake. We organized the references after adding the papers the reviewer mentioned above and corrected the irregularities.

2. How can the authors recycle the objects?

Reply: This is a great suggestion and reminder. Indeed, the recovery of the particles from a single-component 3D printed object is straightforward. The object can be mechanically smashed into a powder and this powder can be used to create a recycled ink with a controllable color response by addition of an adjusted amount of ingredients such as CB and PEO-PPO-PEO copolymer (Pluronic F-108). Multi-material prints can also be recycled with a similar approach but this will yield less color control on the recycled ink. We now briefly mention this possibility in our revised manuscript as shown below.

With a silica **weight** fraction of over 97%, our objects are recyclable and do not display the toxicity issues of typical colored pigments. As opposed to paintings and coatings, chipping of the surface does not compromise its optical functionality, since the color is an intrinsic property of the bulk material. **Easy recycling can be achieved via mechanical pulverization of objects and further addition of CB and PEO-PPO-PEO copolymer to yield a new ink.**

Reviewer #2 (Remarks to the Author):

3D printed patterns of colloidal glasses were experimentally demonstrated for isotropic structural colors, which is never reported. Figures are well organized and manuscript is well written. Therefore, I would like to recommend publication after addressing minor issues.

General reply: We thank the Reviewer for thoroughly reading our paper and for providing helpful comments. We are also glad to hear that she/he finds the content of our paper novel and recommends its publications. The reviewer raises concerns about the practicality of our methods and the angle independence of the color observed. We address these and other remarks of the reviewer below.

- Since this work intended to show the potential of photonic glasses in practical application, authors should demonstrate the various lines of photonic glasses with different diameter and their reluctance spectra should be compared. Authors should discuss whether the diameter of line affect the reflectance spectra. What would be minimum size of line for structural coloration?

Reply: This is indeed a good suggestion but the limitation here is the print line thicknesses that can be created using the DIW method. We could print a minimum line thickness of 150 μm in this work and this is sufficient for coloration. The work of Takeoka and co-workers (doi.org/10.1021/acsnm.0c01366) report that colorful photonic balls down to a size of 20 μm are possible and this size decrease even increases the color saturation. This indicates that such small thicknesses are accessible for coloration. While print line thicknesses down to a few microns have been created using DIW technology, (<https://doi.org/10.1038/428386a>) such low diameters are very challenging to achieved using particle-containing inks. Based on previous work on colloidal aggregates (<https://doi.org/10.1073/pnas.1506272112> and <https://dx.doi.org/10.1364/OE.418735>), nozzle diameters above the 150 μm size used in this work will have no effect on the reflection response.

We now added the following discussion to the revised manuscript.

Although we have used a nozzle diamater of 410 μm in this work, increasing the resolution of the print lines can in principle enhance the saturation of the color or decrease the level of CB needed for a similar color response, as previously suggested.⁴⁹

- Also, in single line on substrate, photonic glasses in line may be dried anisotropically. In other words, only height may be reduced during the drying process, which may affect the angle-dependent color of photonics

glasses. Therefore, cross-sectional SEM or TEM images of photonics glasses should be added after drying and structure of silica particles should be analyzed.

Reply: We thank the reviewer for this intriguing suggestion. We have indeed taken many SEM images of the 3D printed and single-line samples from different angles and cross sections. No obvious sign of structural anisotropy was noticeable in dried samples. Although a 3D printed object will dry anisotropically due to the effect of the substrate, the drying process still leads to structures that show isotropic color response. To confirm this we performed additional reflection measurements (supplementary Figure S5) that prove the angle-independent coloration of our samples (see answer to the question below). While SAXS measurements on the cross-section of the 3D printed samples may shed further light into this matter, this is out of the scope of the current work. The newly added Figure S5 show almost identical reflection spectra at different angles, which indicates structural color isotropy.

- Angle-dependent colors of photonics glass line should be confirmed in reflectance spectra.

Reply: We thank the reviewer for this suggestion. We now performed new experiments to prove the angle independence of the printed structures. In these experiments we performed reflection measurements in which the detection angle was changed from 15° to 65° for all the investigated silica particle sizes. As seen in the new supplementary Figure S5 below, the angle does change the reflection spectra within the noise range. The overlay of spectra obtained at angles of 15°, 40° and 65° are shown for all particle sizes in Figure S5b. The spectra measured at these three angles are almost identical, confirming the angle-independent nature of the structural color. The fact that the intensity of the sample at a given wavelength remains unchanged for different angles also proves that our samples are good diffusers. This highlights the homogeneity of the disorder in the assembled structure. Please see below figure panels and the caption for more information.

Figure S5. Reflection spectra of printed photonic glass structures as a function of the observation angle. (a) Sketch of the used set up. Angle reported in the plots is the angle θ shown in the sketch. (b) Three spectra obtained for angles $\theta = 15, 40$ and 65° were overlaid to demonstrate the angle independence of the color of samples prepared with different silica particle sizes. (c,e,g) Spectra of the photonic glass assembled from (c) 200 nm, (e) 250 nm and (g) 300 nm particles at angles θ varying between 15 and 65° . (d,f,h) Heat maps displaying the effect of the angle (θ) and the wavelength on the log of reflection intensity for samples prepared with (d) 200 nm, (f) 250 nm, and (h) 300 nm silica particles.

We added the following statement in the main text to refer the reader to this new set of results:

“Reflectance measurements at various angles proves the angle-independent nature of the color (Figure S5)”

The nearly identical spectra for angles θ between 15° and 65° proves that the color response of the 3D printed photonic glass samples are angle independent. The fact that the intensity of the reflection at a wavelength remains unchanged at different angles proves the diffusive structure of the photonic glass and also indicates that the structure exhibits no anisotropy due to uneven drying.

- It would be very helpful to show how structural color is developed as a function of the drying time.
Reply: We have now performed new experiments where we quickly heated our disc-shaped sample over a hot plate while imaging the sample with a camera from the side. This new experiment shows the evolution of the sample color over time and temperature (see snapshots below). We observe that the first color change of the sample occurs when the water starts to

evaporate. As a result of initial water removal, the color turns from black to gray. When the initially accessible water is lost, the whole sample becomes gray (3rd image from left). At 150°C the PEO-PPO-PEO copolymer (Pluronic F108) starts to melt and disintegrate. As a result of water and co-polymer removal from the structure, the particles rearrange and densify to form a colloidal glass. The structural color emerges at this stage. At the higher temperature of 200°C, the colloidal glass is consolidated all over the sample. We now inserted this data as supplementary Figure S3 to the paper.

Figure S3. Evolution of the color of the silica structure during heating. The photographs depict a disc-shaped sample made from an ink containing 250 nm silica particles. The sample was subjected to quick heating to demonstrate the emergence of color as the temperature increases. The first color change of the sample is observed when the water starts to evaporate upon heating up to 100°C. Removal of water changes the color from black to gray. At 150°C the PEO-PPO-PEO copolymer (Pluronic F108) starts to melt and disintegrate. As a result of water and copolymer removal from the structure, the particles rearrange and densify to form a colloidal glass. Green structural color emerges at this stage. At the higher temperature of 200°C, the colloidal glass is consolidated all over the sample. Scale bar is 1 cm.

- In line 51, angle-independent scattering should be corrected as 'angle-independent multiple scattering'.

Reply: We thank the Reviewer for this suggestion. We now corrected it in the revised manuscript as suggested by the reviewer.

- In Figure S1, many micron-sized holes are observed. Authors should explain what are those and what caused that.

Reply: Before the 200°C treatment the particles assembly is porous due to the fact that there was still water and PEO-PPO-PEO copolymer present in the printed structure which was occupying space. During the preparation of the sample for SEM, the vacuum applied led to drying of the water from the specimen but prevented the rearrangement of the particles and shrinkage of the assembly, resulting in the observed pores. In contrast, at 200°C for 1 hour the Pluronic F108 copolymer disintegrates and leaves the structure together with the remaining water, therefore rearrangements and densifications occur. As a result of this, the material shrinks and closes the porous structure. This fact we discuss now in more details in the revised text. Please see newly added supplementary Figure S1 on the thermal analyses of the printed parts.

- Is there any mixing between lines with different photonics glasses during drying process?

Reply: Thanks for this interesting question. Due to the viscoelastic nature of the ink, the nanoparticles taking part in the formulation are expected to be jammed and not diffuse after the applied shear is stopped. The elastic recovery test shown in Fig 2c is performed to simulate the printing process and quantify the timescale needed for the ink to restore its solid-like elastic properties. The results indicate that the ink reaches most of its original elastic modulus and solidifies right after the discontinuation of the applied shear. Therefore, the resolution of printing with different colors is expected to be identical to the resolution of the 3D printing setup. To verify this, we imaged the interface between two different colors in one of our printed structures (see below). The reflection microscopy image of the interface of the two different regions nicely demonstrates the sharp interface between these two regions with different colors. The presence of such a sharp interface between the different colored domains printed using the same nozzle diameter (580 μm) is an indication that the resolution of 3D printing is not lowered in our multicomponent printing approach. See below the description and Supplementary Figure we insert now as SI Figure S7.

Figure S7. Interface between the colored domains of a multimaterial printed object. The reflection optical microscopy image (on the right) of the interface between the two differently colored domains demonstrates the sharpness of the printing lines. Such a sharp interface was obtained using the same nozzle diameter of 580 μm , which indicates that the resolution of the 3D printing process is not lowered when multiple inks are used.

- Authors should mention the isotropic structural colors from form factor in introduction including references (Small 15(23), 1900931 (2019) and others).

Reply: Thank you for the suggested reference. In terms of color formation, our system resembles the particle arrangement reported before by Forster et al. (doi.org/10.1002/adma.200903693) and Schertel et al (doi.org/10.1002/adom.201900442). This means that the reflectance intensity is not based solely on the form factor but also has a main contribution from the structure factor. So color arises from the scattering by individual particles but also due to their spatial arrangement. The intensity of reflection in our case is given by $I(\theta) = F(\theta) \cdot S(\theta)$ where $F(\theta)$ is the angle-dependent form factor and $S(\theta)$ is the angle-dependent structure factor. The Figure S1 of Schertel et al.'s paper (doi.org/10.1002/adom.201900442) nicely illustrates the respective contributions of $F(\theta)$ and $S(\theta)$, as shown below

Figure SI.1: Scattering strength λ/ℓ^* plotted versus size ratio r/λ for the ECPA scattering model^[1,2] for $n = 1.6$ and $f = 0.3$ (black line). The same calculation performed with no structural correlation ($S(\theta) = 1$, green line), no single sphere resonance ($F(\theta) = 1$, blue line, scaled by 0.01) and without an effective index influence ($n_{\text{eff}} = 1$, grey dashed line) as well as using the Maxwell-Garnett value for the effective index ($n_{\text{eff}} = n_{\text{MG}}$, grey dashed dotted line).

In our system the color does not solely depend on a single particle property in contrary to the work of Kim et al.⁵³ but originates due to both scattering from a single particle and the interference of scattered waves from the assembly of the particles. Therefore, the color emerges from the form factor in addition to the structure factor of the system. The form factor describes the angle dependence of scattering from individual particles and can be calculated from Mie theory; the structure factor describes the constructive interference of scattered waves via different particles, which can be calculated e.g. from the Percus–Yevick equation⁵⁴. More discussion on such can be found in the following studies^{26,31}.

REVIEWER COMMENTS

Reviewer #1 (Remarks to the Author):

It is glad to see that the authors have made great efforts to improve the quality of this paper. However it would be great to address the following before publication:

1. Reply to “the reply of major question #2”: Before thinking deeply into the explanation of the Figure 3b, one question has to be asked to exclude the effect of any impurities: have you repeated the measurement of TGA for the ink? Is the small peak there real? Also, it seems the PEO-PPO-PEO starts to degrade from 200 °C from Figure S1a. So I will be careful with the discussion about the shift of degradation temperature for PEO-PPO-PEO from 300 to 200 °C. It is better to precisely and properly discuss this point in the paper.

2. How did you make carbon black suspend in aqueous solution? Do they disperse well? Have you made any modification on them? It is supposed to be not suspended in water or organic solvents. Could the authors provide a photo of the ink they used in the supporting or somewhere which seems not appear anywhere in the paper. It is better to give people an impression how it is like. Could you evaluate the homogeneity of carbon black in your ink, for example by SEM?

3. Could the authors provide some SEM images to show how the carbon black is dispersed after the heat treatment?

4. If the drying is changed to heat treatment, it is better to change that in the diagram in Figure 1.

5. Reply to “the reply of major question #4: I am not sure if the other parameters affecting the color of the structure were unchanged. If all of them are the same, how do the authors explain that the times the reflectance peaks are larger than the monodisperse particle size vary from 2.0 to 2.3 as stated in line #241 on page #8.

6. Reply to “the reply of major question #6 and #7: Could the authors provide some photos and spectra for the samples without carbon black for comparison with the ones with carbon black to support their argument of the importance of carbon black. Could they add the corresponding discussion in the manuscript? I believe this will also be useful to support their argument in the reply of major question #9 if that is true.

7. Reply to “the reply of major question #11: Could the authors add some discussion about the resolution and the method to improve it in the paper?

Reviewer #2 (Remarks to the Author):

Authors addressed all the concerns in revised manuscript and response letter. Therefore, I would like to recommend acceptance for publication.

RESPONSE TO REFEREE COMMENTS

Note that we have kept the **yellow highlights** of the previous round and made **green highlights** for the modifications of this round for easy tracking.

REVIEWER COMMENTS

Reviewer #1 (Remarks to the Author):

It is glad to see that the authors have made great efforts to improve the quality of this paper. However it would be great to address the following before publication:

General reply: We thank the Reviewer for thoroughly reading our revised paper and for positive comments about the improved quality. Below, we address all remarks of the reviewer.

1. Reply to “the reply of major question #2”: Before thinking deeply into the explanation of the Figure 3b, one question has to be asked to exclude the effect of any impurities: have you repeated the measurement of TGA for the ink? Is the small peak there real? Also, it seems the PEO-PPO-PEO starts to degrade from 200 °C from Figure S1a. So I will be careful with the discussion about the shift of degradation temperature for PEO-PPO-PEO from 300 to 200 °C. It is better to precisely and properly discuss this point in the paper.

Reply: We thank the reviewer for this reminder. We have indeed repeated the TGA of the ink twice. Indeed, the TGA plot shown in the SI is a repetition of the one in the main text. We observed the same small peak here, therefore, we think that it is real and not due to an impurity. Based on this we made the explanation drafted in your 1st revision.

It is true that the degradation of PEO-PPO-PEO is dependent on the heating parameters such as heating rate, temperature and dwell time. When we hold the temperature at 200°C for prolonged time (see SI Fig. S1f) we observed that almost all PEO-PPO-PEO is degraded. The discussion about the temperature shift is because of the fact that although all other parameters of the experimentation are kept unchanged the temperature of degradation has a clear shift from 300°C to 200°C by simply adding the PEO-PPO-PEO in the ink. And we attribute this to the presence of other ingredients present in the ink.

We have now updated and extended our discussion in the SI as follows.

The slight mass gain observed during the thermogravimetric analysis of the ink (Figure 3b) was confirmed by repeating the TGA experiment (Figure S1d). This small peak is likely due to the uptake/absorbance of oxygen as a result of oxidation of the dried material. Such oxidation occurs right before the copolymer starts to decompose at this temperature.

2. How did you make carbon black suspend in aqueous solution? Do they disperse well? Have you made any modification on them? It is supposed to be not suspended in water or organic solvents. Could the authors provide a photo of the ink they used in the supporting or somewhere which seems not appear anywhere in the paper. It is better to give people an impression how it is like. Could you evaluate the homogeneity of carbon black in your ink, for example by SEM?

Reply: We thank the reviewer for this comment. Carbon black (CB) is normally soluble in organic solvents and not in aqueous solutions. The presence of the surfactant PEO-PPO-PEO promotes the dispersion of the CB particles. Aqueous CB suspensions can readily be prepared with PEO-PPO-PEO concentrations of 2% or higher (<https://onlinelibrary.wiley.com/doi/10.1002/adma.201000356>). In our ink we have excess PEO-PPO-PEO to sterically disperse the CB well. Therefore, CB dispersion was straightforward for our ink preparation. We agree that having a photo of the ink may help the reader. We now added this to the Supplementary Information as Fig. S9.

Figure S9. Photographs of the ink containing carbon black. (a) Highly homogenous ink obtained after the 3-roll milling process. (b) Ink manipulated with a spatula. The inset shows the 3D printed grid before the heat treatment. Scale bar in the inset is 5 mm.

The newly added Supplementary Figure S10 shown below (and homogenous pictures of the inks given above, Figure S9) indicates that the carbon black is fairly homogeneously distributed in the ink. Figure S10 shows the SEM images of the printed sample after heat treatment. Here we assume that the small particles next to the spherical silica colloids to be mostly the carbon black content in the system.

Figure S10. SEM micrographs of the printed structure after the heat treatment. (a,b) Images obtained at the magnifications of (a) 14k and (b) 48k. The homogenous distribution of the carbon black particles through the ink is illustrated using the red marks. Since the CB shows an average size of 7 nm, most of the particles are not visible in these images.

3. Could the authors provide some SEM images to show how the carbon black is dispersed after the heat treatment?

Reply: Please refer to the previous response to comment #2. We have now added a new Supplementary Figure S10 where we present some SEM micrographs after the heat treatment step and highlight the non-spherical content with red marks to evaluate the distribution of the CB particles. Note here that the single CB nanoparticles (< 10 nm) expected in the printed samples are too small to be visualized in the SEM image.

4. If the drying is changed to heat treatment, it is better to change that in the diagram in Figure 1.

Reply: We thank the reviewer for this reminder. We have now updated the Figure 1 as suggested.

5. Reply to “the reply of major question #4: I am not sure if the other parameters affecting the color of the structure were unchanged. If all of them are the same, how do the authors explain that the times the reflectance peaks are larger than the monodisperse particle size vary from 2.0 to 2.3 as stated in line #241 on page #8.

Reply: We thank the reviewer for this comment and careful thinking and reading. The estimate of the reflection peak from the particle size to $\lambda = 2.3 d$ is approximate, it depends on the effective refractive index but also on the shape of the spectrum, which is influenced by the structure but also the individual particles' form factor. The calculation

based on the structure factor, $Q_{peak} \sim 2.3 \frac{\pi}{d}$ means $\lambda_{peak} \sim \frac{4}{2.3} n_{eff} d = 2.22 d$ for $n_{eff}=1.28$. Here, λ_{peak} is proportional to n_{eff} which depends on the volume fraction of particles that may lie between 0.5-0.65. Moreover, the peaks are not really sharp and the wavelength dependence of the background diffuse scattering, as well as contributions from the particle form factor, influence the reflection spectra which can lead to subtle shifts of the shallow maximum.

We have rephrased our statement on page #9 and now write:

From the above analysis, we expect a peak in single scattering back-reflection to occur at $\lambda_{peak} \sim \frac{4}{2.3} n_{eff} d$. For $d = 250\text{nm}$, we obtain a peak wavelength (λ_{peak}) of 557 nm, which is in good agreement with the peak at 535 nm obtained from the measurements on objects made with 250 nm silica particles (Figure 4d). When not directly reflected, single scattered light is either lost (absorbed or transmitted) or multiply scattered^{27,32}. In our optically thick samples (filament diameter 0.58mm), we can neglect the transmission pathway. Multiple scattering leads to an undesirable diffuse broadband background which we suppress by adding the broadband absorber carbon black.

The predicted reflectance peak wavelengths (Figure 4d) are about 2.2 times larger than the monodisperse particles size (d), thus providing a simple guideline for the selection of the colloidal particles to be added to the ink depending on the desired structural color. Slight shifts in the peak wavelength can be explained by variations in the solid volume fraction and the relatively broad nature of the spectra, which arises from residual diffusive scattering from the individual particles^{27,50}.

In addition, we have made the following adjustment and noted that volume fraction of the inks slightly varies depending on the batch:

The other parameters expected to influence the color of the structure, such as the volume fraction of the particles, the refractive index of the medium and the amount of CB, were kept unchanged in this demonstration except the volume fraction. The volume fraction of the colloids in the ink may slightly vary during ink homogenization (see Supplementary Information for details).

SI addition:

Influence of the 3-roll milling step on the colloid volume fraction in the ink:

We have used a 3-roll mill to homogenize the ink and break down particle aggregates that could cause nozzle clogging during DIW printing. During the 3-roll milling process, the ink is exposed to air and it is spread on the high-surface-area rolling cylinders. This leads to water evaporation and lowers the water

content of the ink. Therefore, it is not easy to control the water content of the ink during this homogenization step. As a result of this evaporation effect, the final solid volume fraction and the distance between colloids after the 3D printing and heat treatment steps may not match closely the predicted values. This affects directly the reflectance of the printed object as shown in earlier work [26].

6. Reply to “the reply of major question #6 and #7: Could the authors provide some photos and spectra for the samples without carbon black for comparison with the ones with carbon black to support their argument of the importance of carbon black. Could they add the corresponding discussion in the manuscript? I believe this will also be useful to support their argument in the reply of major question #9 if that is true.

Reply: We now added photos and reflectance spectra of the printed samples with and without CB to demonstrate the influence of the CB addition on the color and the reflectance of the samples. This new data clearly shows that the presence of CB is necessary for the color saturation in our samples. The new Supplementary Figure S10 and the discussion is shown below.

Figure S6. Comparison of printed samples with and without the addition of carbon black (CB). (a,b) Photographs of a printed sample prepared with 250 nm sized particles containing (a) 1.4 % CB and (b) no CB. (c) Reflectance spectra of 3D printed samples with and without CB. The presence of carbon black in the 3D printed sample allows for the emergence of the green color. Upon heat treatment of this same sample at 850°C for 3 hours, the CB is removed and the color turns to white with stronger reflectance. This suggests that the CB inhibits multiple scattering events and thus saturates the color.

7. Reply to “the reply of major question #11: Could the authors add some discussion about the resolution and the method to improve it in the paper?”

Reply: We have now added the following discussion on the resolution of the 3D prints and how to improve on that end in the Supplementary Information and refer to that in the Materials and Methods section.

Resolution of the DIW 3D printing method:

The resolution of the DIW method usually lies around 100 μm , but can be extended down to a few microns by printing the ink in a liquid at high extrusion pressures⁵⁵. Here, we used a 410 μm nozzle to achieve sufficient resolution without causing clogging during the printing process. DIW of particle-containing inks below 100 μm resolution is challenging, since clogging might become an issue for nozzles below this size. This originates from the fact that our ink has a high concentration of particles. Therefore, any particle aggregates in the suspension have to be disintegrated within the ink before printing. To this end, the particle suspensions were subjected to a 3-roll milling process, which reproducibly led to homogenous inks that can be flawlessly ejected from 410 μm nozzles. In principle, ball-milling the suspensions over long hours may help to further remove aggregates and possibly improve the resolution of the DIW process. Alternatively, the resolution can be enhanced by developing inks that can be printed using stereolithography techniques⁵⁶.

Reviewer #2 (Remarks to the Author):

Authors addressed all the concerns in revised manuscript and response letter. Therefore, I would like to recommend acceptance for publication.

Reply: We thank the reviewer for taking the time to review our paper and providing useful, detailed comments which improved the overall quality of the work.

REVIEWERS' COMMENTS

Reviewer #1 (Remarks to the Author):

The authors addressed all the concerns. Therefore, I would like to recommend to publication with no further revision